# Isolation and Characterization of Starch from Different Potato Cultivars Grown in Croatia

Mario Kovač [1], Boris Ravnjak [2,*], Drago Šubarić [3], Tomislav Vinković [2], Jurislav Babić [3], Đurđica Ačkar [3], Ante Lončarić [3], Antonija Šarić [3], Vesna Ocelić Bulatović [4] and Antun Jozinović [3]

1   Faculty of Agriculture and Food Technology, University of Mostar, Biskupa Čule bb,
    88000 Mostar, Bosnia and Herzegovina; mario.kovac11@gmail.com
2   Faculty of Agrobiotechnical Sciences Osijek, Josip Juraj Strossmayer University of Osijek, Vladimira Preloga 1,
    31000 Osijek, Croatia; tvinkovic@fazos.hr
3   Faculty of Food Technology Osijek, Josip Juraj Strossmayer University of Osijek, Franje Kuhača 18,
    31000 Osijek, Croatia; dsubaric@ptfos.hr (D.Š.); jbabic@ptfos.hr (J.B.); dackar@ptfos.hr (D.A.);
    ante.loncaric@ptfos.hr (A.L.); antonija.saric@ptfos.hr (A.Š.); ajozinovic@ptfos.hr (A.J.)
4   Faculty of Chemical Engineering and Technology, University of Zagreb, Marulićev trg 19,
    10000 Zagreb, Croatia; vocelicbulatovic@fkit.hr
*   Correspondence: bravnjak@fazos.hr

**Abstract:** Starch is a polysaccharide that is widely used in food and other industries; therefore, due to its great potential, it is attempted to be maximally isolated from various foods rich in starch. Commonly, potatoes are used for starch production due to the relatively high starch content in tubers, and the process itself is complex and includes several steps. The aim of this study was to isolate and characterize starch from eight potato varieties. First, the basic chemical composition of the potato samples was determined, and then the isolation was carried out under laboratory conditions. The isolated starch was air dried, then ground and sieved through a 400 μm sieve. The basic chemical composition, amylose content, starch color, swelling capacity and solubility index, clarity of starch pastes, texture of starch gels and thermo-physical properties (gelatinization and retrogradation) were determined in the obtained starch samples. The results showed that the SL 13–25 potato variety had the lowest starch content, while the Stilleto variety had the highest starch content. The content of protein, fat, ash and crude fiber was relatively low in all of the isolated starches, indicating their high purity. Also, the difference in the color of the isolated starches was difficult to see. The highest amylose content had starch from Saprodi, while the lowest was observed in starch from SL 13–25. Starch from the cv. Dartiest had the highest, while starch from the cv. Sereno had the lowest gel strength. The starch of the cv. Dartiest also had the highest clarity value. The retrogradation transition temperatures and enthalpies were lower compared to the gelatinization temperatures and enthalpies. With the increasing temperature, both the swelling capacity and the solubility index of all the samples increased. According to the obtained differences, this study confirmed the significant influence and role of different cultivars on starch characteristics.

**Keywords:** potato; starch; isolation; characterization

## 1. Introduction

Potatoes are the fourth most important food crop in the world after wheat, rice and maize, with a global annual production of approximately 300 million tons. The chemical composition and ratio of nutrients differ depending on the potato cultivar, but also on a whole series of ecological and production factors (soil, fertilization, climatic conditions, etc.) and the method of preparation [1–3].

Potato is a raw material rich in starch, which is found in the potato tuber and is formed by dividing the ends of the stolons, after which the cell swells and the tuber develops [4]. Potato starch is an essential product obtained through the process of the separation of the

solid components—starch and fiber—from the liquid potato juice. The goal of the starch production industry is to obtain starch with as few impurities as possible—i.e., to obtain the highest yield of starch—and this is a complex and demanding process that includes a series of steps [5,6].

The process of producing starch from potatoes can be conducted in three ways; namely, by separating the three components during production: the starch itself, fibers and potato juice. The economy over a long period of processing should also be taken into account, as well as parameters such as the dry matter, starch and protein content [7].

This research aimed to examine the influence of different potato cultivars on the starch properties: starch color, swelling capacity and solubility index, clarity of starch pastes, texture of starch gels and thermophysical properties of starch (gelatinization and retrogradation). For this purpose, starch was isolated from eight potato cultivars grown in the area of Osijek-Baranja County.

## 2. Materials and Methods

### 2.1. Materials

Eight different starchy potato cultivars (Saprodi, Sofista, Stilleto, Dartiest, Sereno, Senata, Scala and future cultivar (breeding clone) with code name SL 13–25) were grown in 2021 at the Experimental Station of the Faculty of Agrobiotechnical Sciences in Osijek, Croatia, located near the suburban settlement Tenja (45.518770, 18.774791), Osijek-Baranja County. Starchy potato cultivars were obtained from starch potato seeds producer Royal ZAP-Semagri Holland BV (Wieringerwerf, The Netherlands). Potato seeds were delivered to the faculty to test the quality and possibility of their production in the area of Eastern Europe (temperate to dry climate). Planting was performed mechanically on 3 May 2021 (plant spacing $32 \times 70$ cm; plant density 4.46 plants per square meter). The trial was set in a randomized block design, where each cultivar was planted in four repetitions, each containing 100 plants. For sampling, tuber harvesting was performed manually on 29 September 2021. After the harvest, potato tubers were stored for three months at 4 °C, and this was the time of tuber maturation. Afterwards, laboratory analysis and starch isolation were performed.

### 2.2. Method of Isolation of Starch from Potatoes

The process of isolating starch from potatoes began with a thorough washing of the potato tubers in tap water, after which the potato tubers were washed in distilled water. The potatoes were then cut into small pieces and ground using a laboratory blender (Kenwood) with distilled water (0.5 kg potatoes/250 mL water) for about 25 s (at speed level 5). The next step was to manually wash the starch from the potato pulp. Potato pulp was poured onto starch gauze with the symbol 11xxx [132 μm] and the starch was rinsed with distilled water. The process of washing the starch was repeated 4–5 times until the starch was no longer slippery and the water became clear. The obtained starch milk was left overnight (12 h at room temperature) for starch sedimentation, and then the water was poured off over the sediment (decantation). Then, the starch was mixed with a new portion of distilled water (the process took place in a bucket, and the amount of distilled water was 5 L). This operation was performed two more times (starch was rinsed 3 times in total). In the fourth mixing of starch with distilled water, refining was performed—i.e., a suspension of starch and distilled water was poured through 13xxx [104 μm] starch gauze—and the starch was allowed to sediment again (12 h). After the final pouring of water over the precipitate, the wet starch was placed in plastic trays to dry. The samples were dried in a laboratory dryer, Memmert UFE 500, Schwabach, Germany, at a temperature of 30 °C. The dried starch was ground in an IKA M20 laboratory grinder, Staufen, Germany, with a grinding time of 6 s, and then sieved through a sieve with an opening size of 400 μm. The starches were packed in polyethylene bags and then their properties were determined.

### 2.3. Determination of Basic Chemical Composition and Amylose Content

The chemical composition was determined according to the following ISO standards: determination of dry matter content through drying to constant mass at 105 °C (ISO 6540), determination of ash content through burning the sample in a muffle furnace at 550 °C (ISO 5984), determination of fat content using the Soxhlet method (ISO 6492), determination of protein content using the Kjeldahl method (ISO 5983-2), determination of crude fiber content using the method with intermediate filtration (ISO 6865). Amylose content was determined according to the Megazyme method using a commercial amylose/amylopectin enzymatic kit K-AMYL 06/18 (Megazyme Ireland International, Ltd., Bray, Ireland).

### 2.4. Determination of the Color of Starch

Chroma Meter CR-300, Konica Minolta, Tokyo, Japanwas used to determine the color of the starch. Before measuring the color in the CIELab and CIELCh systems (L*—black/white; a*—red/green; b*—yellow/blue; C—chroma, h—hue), the device was calibrated using a calibration plate. Five measurements were performed for each sample.

### 2.5. Determination of Paste Properties by Brabender's Micro Visco-Amylograph

Determination of paste properties was carried out using Brabender's micro visco-amylograph (Brabender, Duisburg, Germany). The device is connected to a computer that manages the operation of the device and processes the obtained results. When measuring the rheological properties, the samples were subjected to a temperature program:

1. Heating from 30 to 92 °C, heating rate 7.5 °C/min.
2. Isothermal at 92 °C, 15 min.
3. Cooling from 92 to 50 °C, cooling rate 7.5 °C/min.
4. Isothermal at 50 °C, 15 min.

### 2.6. Determination of Swelling Power and Solubility Index of Starches

The isolated starches were weighed in 50 mL centrifuge cuvettes with a conical bottom to prepare a 1.0% suspension (25 g suspension). The cuvettes were covered with a cap and mixed well, and the suspension was then heated at temperatures of 55, 65, 75 and 85 °C for 30 min in a water bath with a shaker (150 rpm), with occasional manual mixing to homogenize the sample. Gelatinized starch was then cooled as soon as possible in cold water to room temperature and centrifuged at 4000 rpm for 15 min. After centrifugation, the supernatant was decanted into pre-dried and weighed aluminium drying containers and dried at 105 °C to a constant mass.

### 2.7. Determination of Clarity of Starch Pastes

Paste clarity of the starches was determined based on the method of Kerr and Cleveland (1959). The calculated mass of starch and distilled water up to a mass of 20 g was added to previously weighed empty cuvettes, and the samples were then homogenized on a Vortex shaker. A 1% suspension of each sample was heated in a boiling water bath for 30 min with constant stirring. The cuvettes with the samples were then removed from the bath and allowed to cool at room temperature for 1 hour. If there is a loss in the mass of the suspension, it is compensated by the addition of water that has evaporated, and the mixing on the Vortex is repeated. Finally, the samples were measured for transmittance at 650 nm using a laboratory LLG Uni Spec 2 spectrophotometer.

### 2.8. Determination of the Texture of Starch Gels

The TA-XT Plus device, Stable microsystem, UK, was used to determine the textural properties of the samples. The sample was analyzed after 24 h to remove air bubbles incorporated during the preparation of the gels. The gel structure analysis procedure is carried out using a cylindrical probe with the following dimensions: diameter = 25 mm; height = 35 mm.

Measuring method:

1. Speed before measuring—3 mm/s;
2. Speed of measuring (during penetration)—2 mm/s;
3. Speed after measuring—10 mm/s.

The penetration depth of the measuring body during measurement is 20 mm, and the measuring force is 1 g.

### 2.9. Determination of Thermophysical Properties

To determine the thermophysical properties of the samples, a Mettler-Toledo DSC model 822e differential scanning calorimeter (DSC) (Mettler Toledo, Giessen, Germany) was used, and the measurements were carried out in a nitrogen atmosphere of purity 5.0 (Linde). The results were displayed and processed using the STARe Evaluation V6_V12 Conversation software. The control or verification (so-called check) of the reliability of the instrument (module), which determines the difference between the measured and determined reference value of the temperature or heat flow of the tested reference material, was carried out using indium. Cooling was carried out using liquid nitrogen (100 L container, Messer, Frankfurt, Germany).

### 2.10. Statistical Analysis

Data presented in this work are expressed as the mean value $\pm$ SD (standard deviation) from three parallel repetitions for all analysis. One-way analysis of variance (ANOVA) and Duncan's least significant difference (LSD) test were used (Statistica 13.5, TIBCO Software Inc., Palo Alto, CA, USA) to assess the significance of the differences between the mean values at a significant level of $p < 0.05$.

## 3. Results and Discussion

### 3.1. Basic Chemical Composition of Tested Potato Cultivars

The chemical composition of potatoes is very important for the potato starch industry. The most important parameter is the amount of starch present in the potato so that at the end of the process, the economy and utilization are as high as possible. The type and cultivar of the potato itself, as well as the growing conditions (soil, climate, fertilizer addition, etc.), should also be considered; for instance, potatoes grown on silty soil have the highest, while those grown on sandy soil have the lowest proportion of dry matter [8]. For this research, starches were isolated from eight potato cultivars, whose properties are shown in Table 1. As Table 1 shows, the Stilleto cultivar had the highest proportion of dry matter, and thus the highest proportion of starch, which could have been assumed as most of the dry matter consists of starch. The Senata cultivar had the lowest proportion of dry matter. However, the lowest proportion of starch was found in the SL 13–25 cultivar. The content of dry matter in potatoes is a very important factor, with an average value of 22.3%, and can vary significantly depending on the cultivar, conditions during growth and the degree of maturity [7]. Changes also occur during storage itself. The share of water in all of the cultivars was above 78.34%. Grommers and van der Krogt [7] state that, on average, water makes up more than 70% of the total mass of potatoes, which was confirmed by the above results. Potatoes used for processing into starch must be healthy, must not sprout and must contain at least 14% starch [9]. In this research, the proportion of starch ranged between 17.26 $\pm$ 0.37 and 22.49 $\pm$ 0.71%, which confirms the suitability of these cultivars for starch production. The proportion of protein ranged between 1.79 $\pm$ 0.09 to 3.47 and 0.16%. The Stilleto cultivar had the highest protein content, while Senata had the lowest. The highest proportion of fat was found in the SL 13–25 cultivar, and the lowest proportion in the Senata cultivar. The Sereno cultivar had the highest, while the Sofista cultivar had the lowest share of crude fiber.

**Table 1.** Chemical composition of the investigated potato cultivars.

| Sample | Dry Matter [%] | Protein [%] | Fat [%] | Ash [%] | Starch [%] | Crude Fiber [%] |
|---|---|---|---|---|---|---|
| Saprodi | 25.05 ± 0.29 [c] | 2.53 ± 0.11 [c] | 0.09 ± 0.01 [a,b] | 1.24 ± 0.01 [d,e] | 19.78 ± 0.17 [b] | 1.08 ± 0.01 [d] |
| Sofista | 24.43 ± 1.99 [b,c] | 2.15 ± 0.13 [b] | 0.09 ± 0.00 [a,b] | 1.24 ± 0.01 [d,e] | 19.91 ± 0.12 [b] | 0.88 ± 0.04 [a] |
| Stilleto | 28.54 ± 0.69 [d] | 3.47 ± 0.16 [d] | 0.10 ± 0.00 [b] | 1.02 ± 0.01 [b] | 22.49 ± 0.71 [c] | 0.96 ± 0.00 [b,c] |
| Dartiest | 21.95 ± 1.14 [a,b] | 2.37 ± 0.06 [b,c] | 0.10 ± 0.00 [b] | 0.96 ± 0.01 [a] | 17.42 ± 0.00 [a] | 1.08 ± 0.01 [d] |
| Sereno | 24.48 ± 0.76 [b,c] | 3.37 ± 0.03 [d] | 0.09 ± 0.00 [a] | 1.20 ± 0.01 [d] | 17.63 ± 0.43 [a] | 1.18 ± 0.04 [e] |
| Senata | 21.66 ± 0.97 [a] | 1.79 ± 0.09 [a] | 0.08 ± 0.00 [a] | 1.11 ± 0.01 [c] | 17.36 ± 0.32 [a] | 1.07 ± 0.05 [d] |
| Scala | 23.57 ± 0.84 [a,b,c] | 2.15 ± 0.08 [b] | 0.12 ± 0.00 [c] | 1.14 ± 0.02 [c] | 19.06 ± 0.70 [b] | 0.90 ± 0.03 [a,b] |
| SL 13–25 | 23.48 ± 0.72 [a,b,c] | 2.33 ± 0.09 [b,c] | 0.16 ± 0.00 [d] | 1.25 ± 0.03 [e] | 17.26 ± 0.37 [a] | 1.03 ± 0.04 [c,d] |

Values with different letters in the same column are significantly different at $p < 0.05$.

### 3.2. Basic Chemical Composition of Isolated Potato Starches

Potato cultivars differ in the proportion of starch they possess, and the proportion of starch itself defines their applicability in nutrition [10]. Starches were isolated from eight potato cultivars, listed in Table 1, and their chemical composition is shown in Table 2. As stated by Martínez et al. [11], the chemical composition of starches depends on the potato cultivar, as well as on the starch isolation process.

**Table 2.** Chemical composition of potato starch obtained from the tested probes.

| Sample | Dry Matter [%] | Protein [%] | Fat [%] | Ash [%] | Starch [%] | Crude Fiber [%] | Amylose [%] |
|---|---|---|---|---|---|---|---|
| Saprodi | 84.66 ± 0.02 [c] | 0.09 ± 0.00 [a] | 0.01 ± 0.00 [a] | 0.22 ± 0.00 [a,b] | 82.67 ± 0.27 [b] | 1.01 ± 0.03 [d] | 22.42 ± 0.67 [f] |
| Sofista | 85.33 ± 0.02 [e] | 0.10 ± 0.01 [a] | 0.01 ± 0.00 [a] | 0.26 ± 0.00 [c] | 83.25 ± 0.41 [b,c] | 0.77 ± 0.01 [b] | 20.67 ± 0.61 [c,d] |
| Stilleto | 84.67 ± 0.01 [c] | 0.09 ± 0.01 [a] | 0.01 ± 0.00 [a] | 0.22 ± 0.01 [a,b] | 82.96 ± 0.06 [b,c] | 0.95 ± 0.04 [c,d] | 21.16 ± 0.29 [d,e] |
| Dartiest | 85.69 ± 0.02 [f] | 0.10 ± 0.00 [a] | 0.01 ± 0.00 [a] | 0.21 ± 0.00 [a,b] | 83.75 ± 0.13 [c] | 0.70 ± 0.01 [a] | 20.28 ± 0.37 [b,c] |
| Sereno | 84.33 ± 0.08 [b] | 0.15 ± 0.03 [b] | 0.01 ± 0.00 [a] | 0.23 ± 0.01 [b] | 82.51 ± 0.40 [b] | 0.70 ± 0.02 [a] | 21.40 ± 0.48 [e] |
| Senata | 84.92 ± 0.03 [d] | 0.10 ± 0.01 [a] | 0.01 ± 0.00 [a] | 0.26 ± 0.00 [c] | 83.32 ± 0.58 [b,c] | 0.90 ± 0.01 [c] | 19.93 ± 0.34 [b] |
| Scala | 84.95 ± 0.01 [d] | 0.09 ± 0.00 [a] | 0.01 ± 0.00 [a] | 0.25 ± 0.00 [c] | 82.95 ± 0.42 [b,c] | 0.90 ± 0.01 [c] | 16.97 ± 0.37 [a] |
| SL 13–25 | 83.51 ± 0.13 [a] | 0.09 ± 0.00 [a] | 0.01 ± 0.00 [a] | 0.21 ± 0.01 [a] | 81.23 ± 0.10 [a] | 1.02 ± 0.05 [d] | 16.82 ± 0.34 [a] |

Values with different letters in the same column are significantly different at $p < 0.05$.

The protein content of the isolated potato starches in this study varied between 0.09 and 0.15%, and these values are very low compared to the protein values in the starches isolated from three autochthonous potatoes of the Andean region [11], which indicates high purity [12]. The proportion of fat is the same in all of the starches and amounts to 0.01%, and this proportion is lower than that in the Mexican potato starch [13] and potato starch from Venezuela [14]. The ash content ranged between 0.21 and 0.26%, which is similar to the values for the starches from autochthonous potatoes of the Andean region [11] and potato starch from Venezuela [14]. The raw fiber content ranged between 0.70 and 1.02%, while the amylose content ranged from 16.82% in SL 13–25 cultivar to 22.42% in Saprodi cultivar.

### 3.3. The Color of Potato Starches

Starch color is an important parameter of starch quality. Table 3 shows the color parameters of the isolated potato starches. In the CIELab system, the intensity of white color—i.e., the brightness of starch—is expressed through the L* value. The values of the L* parameter range from 0 (black) to 100 (white), and the values obtained for all of the starches were close to 100, which indicates a white color of high intensity. The parameter a* describes the color domain; if the value is positive, it is in the domain of red, and if the value is negative, it is in the domain of green. As the table shows, all of the obtained values are negative, which means that the samples are in the domain of green.

**Table 3.** Color properties of the isolated potato starches.

| Sample | L* | a* | b* | C | h° |
|---|---|---|---|---|---|
| Saprodi | 94.85 ± 0.31 [c] | −1.45 ± 0.19 [a] | 2.29 ± 0.12 [a] | 2.72 ± 0.05 [b,c,d] | 122.36 ± 4.59 [e] |
| Sofista | 94.08 ± 0.45 [a,b] | −1.32 ± 0.02 [a,b] | 2.33 ± 0.07 [a] | 2.67 ± 0.06 [b,c] | 119.46 ± 0.88 [d,e] |
| Stilleto | 93.71 ± 0.17 [a] | −1.21 ± 0.02 [b] | 2.49 ± 0.04 [b] | 2.77 ± 0.04 [c,d] | 115.94 ± 0.59 [b,c,d] |
| Dartiest | 94.49 ± 0.30 [b,c] | −1.15 ± 0.29 [b] | 2.52 ± 0.14 [b] | 2.79 ± 0.14 [d] | 114.40 ± 6.06 [b,c] |
| Sereno | 93.81 ± 0.56 [a,b] | −1.31 ± 0.04 [a,b] | 2.65 ± 0.01 [c] | 2.96 ± 0.02 [e] | 116.35 ± 0.75 [c,d] |
| Senata | 93.67 ± 1.19 [a] | −1.18 ± 0.26 [b] | 2.93 ± 0.05 [d] | 3.17 ± 0.11 [f] | 111.86 ± 4.25 [a,b] |
| Scala | 94.18 ± 0.25 [a,b,c] | −1.32 ± 0.02 [a,b] | 2.31 ± 0.07 [a] | 2.65 ± 0.05 [b] | 119.77 ± 0.96 [d,e] |
| SL 13–25 | 94.85 ± 0.04 [c] | −0.79 ± 0.03 [c] | 2.27 ± 0.04 [a] | 2.41 ± 0.04 [a] | 109.16 ± 0.51 [a] |

Values with different letters in the same column are significantly different at $p < 0.05$.

From the obtained results, it is evident that the values of the parameter b* of all the tested samples are positive, which means that all of the samples are in the yellow domain. Parameter C expresses color saturation, and the highest value, as well as saturation, was in the starch obtained from the potatoes of the Senata cultivar. The h° parameter represents the color tone and is expressed in °, where 0° indicates red, values over 90° indicate yellow, over 180° indicate green and values up to 270° indicate a blue tone.

According to the obtained results, shown in the table, it can be seen that all of the samples are in the yellow domain. Compared to the results of Sit et al. [15], the results for the L* value were slightly lower. Pérez Sira and Amaiz [16] state that L* values above 90 are satisfactory and confirm the purity of the starch. In general, high L* values and low values of the a* and b* parameters confirm that starch can be used in products that require a clean and uniform color.

### 3.4. Paste Properties by Brabender's Micro Visco-Amylograph

Among all commercial starches, potato starch has the highest swelling power and provides the highest viscosity of starch pasting properties [17]. Matvejev et al. [18] concluded that the thermodynamic melting properties of starches are directly correlated with the amylose content. The pasting properties of starch are usually determined based on changes in the starch viscosity. The results of the properties of the pastes determined using Brabender's micro visco-amylograph are shown in Table 4. The highest peak viscosity value was found in the starch sample obtained from the Scala cultivar 1673.5 ± 20.5 BU, while the lowest value was in the starch sample obtained from the Stilleto cultivar 1410.0 ± 2.8 BU. After shearing at 92 °C, the viscosity of all eight potato cultivars decreased because the structure of the paste was damaged due to shearing. Cooling to 50 °C showed an increase in the viscosity of the potato starch paste.

**Table 4.** Pasting properties of the isolated potato starches.

| Sample | Peak Viscosity [BU] | Viscosity at 92 °C [BU] | Viscosity after Shearing at 92 °C [BU] | Viscosity at 50 °C [BU] | Viscosity after Shearing at 50 °C [BU] | Breakdown [BU] | Setback [BU] |
|---|---|---|---|---|---|---|---|
| Saprodi | 1505.5 ± 33.2 [b] | 1130.5 ± 6.4 [f] | 600.0 ± 2.8 [f] | 1046.0 ± 18.4 [e] | 854.5 ± 13.4 [e] | 905.5 ± 30.4 [a] | 446.0 ± 15.6 [d] |
| Sofista | 1499.5 ± 19.1 [b] | 1016.0 ± 7.1 [d] | 539.0 ± 8.5 [d] | 924.5 ± 3.5 [c] | 758.5 ± 0.7 [c] | 960.5 ± 27.6 [b,c] | 385.5 ± 4.9 [b,c] |
| Stilleto | 1410.0 ± 2.8 [a] | 1006.0 ± 18.4 [d] | 514.5 ± 12.0 [c] | 898.0 ± 18.4 [c] | 745.5 ± 33.2 [c] | 895.5 ± 9.2 [a] | 383.5 ± 6.4 [b,c] |
| Dartiest | 1446.5 ± 9.2 [a] | 904.5 ± 16.3 [b] | 468.0 ± 7.1 [a] | 834.5 ± 19.1 [b] | 693.0 ± 11.3 [b] | 978.5 ± 2.1 [b,c] | 366.5 ± 12.0 [b] |
| Sereno | 1518.5 ± 21.9 [b] | 1048.0 ± 11.3 [e] | 562.5 ± 6.4 [e] | 962.5 ± 19.1 [d] | 799.0 ± 15.6 [d] | 956.0 ± 28.3 [b] | 400.0 ± 12.7 [c] |
| Senata | 1490.0 ± 1.4 [b] | 888.5 ± 9.2 [a,b] | 487.0 ± 4.2 [b] | 806.5 ± 6.4 [a,b] | 641.5 ± 10.6 [a] | 1003.0 ± 5.7 [c,d] | 319.5 ± 2.1 [a] |
| Scala | 1673.5 ± 20.5 [c] | 974.0 ± 9.9 [c] | 521.0 ± 4.2 [c] | 915.0 ± 4.2 [c] | 738.0 ± 4.2 [c] | 1152.5 ± 16.3 [e] | 394.0 ± 0.0 [c] |
| SL 13–25 | 1519.0 ± 9.9 [b] | 870.0 ± 9.9 [a] | 474.0 ± 5.7 [a,b] | 796.0 ± 0.0 [a] | 663.0 ± 7.1 [a,b] | 1045.0 ± 4.2 [d] | 322.0 ± 5.7 [a] |

Values with different letters in the same column are significantly different at $p < 0.05$.

According to Chung et al. [19], the increase in viscosity during cooling occurs due to retrogradation—that is, the connection of dissolved starch molecules. Breakdown, which indicates the stability of the starch paste during shearing at high temperatures, is calculated

from the difference in the viscosity value after 15 min of mixing at 92 °C, and the peak viscosity value. The Scala cultivar had the highest breakdown value, which indicates that it is less stable during shearing at high temperatures. The cultivars Senata and SL 13–25 had similar breakdown values. From the setback results, which indicate the tendency of starch paste to retrograde, it is noticeable that the starch obtained from the Saprodi cultivar is slightly more susceptible to retrogradation compared to the other starches.

### 3.5. Swelling Power and Solubility Index

The swelling power and the solubility index show the interconnection of starch chains within the crystalline and amorphous regions of the starch granule, which is determined by the ratio of amylopectin and amylose, their conformation and the degree of branching [20].

Figures 1 and 2 show the swelling power and solubility index of the starches isolated from different potato cultivars at temperatures of between 55 °C and 85 °C. As suggested by the results in the figures, there is a distinct tendency to increase the swelling power, as well as the solubility index, with an increase in the treatment temperature. At the highest temperature, the starch of the Sereno cultivar showed the highest, and the starch of the SL 13–25 cultivar showed the lowest value of swelling power, while the highest value of the solubility index was found in the starch of the Stilleto cultivar, and the lowest in the Senata cultivar. The swelling power and solubility index depend on the type of starch and the ratio of amylose and amylopectin. The values of the swelling power decrease with an increase in the amylose content, and as the amylopectin content increases, the swelling power increases too, and such starch is more stable to freeze-thaw processes, which significantly accelerates the processes of retrogradation and syneresis. Potato starch consists of relatively large granules that have relatively high swelling power values [21]. The morphological characteristics of the granules can also influence the swelling power and solubility index [22].

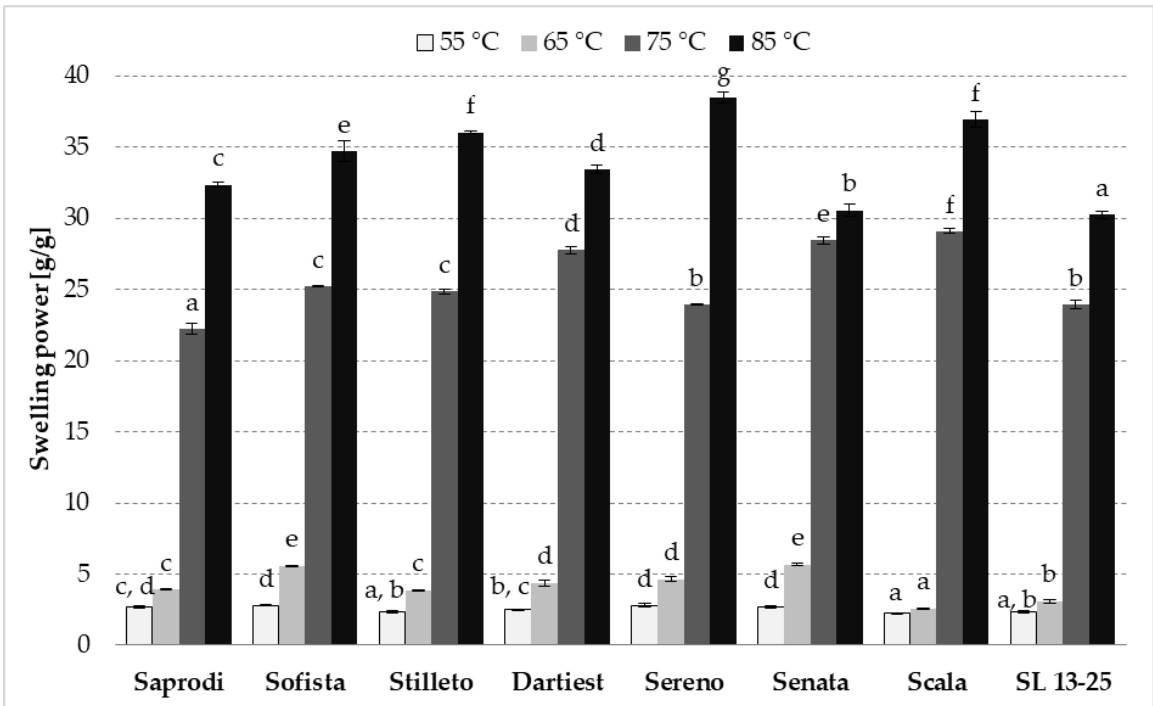

**Figure 1.** Swelling power of the isolated potato starches at different temperatures. Different letters above the bars indicate that the values are significantly different at $p < 0.05$.

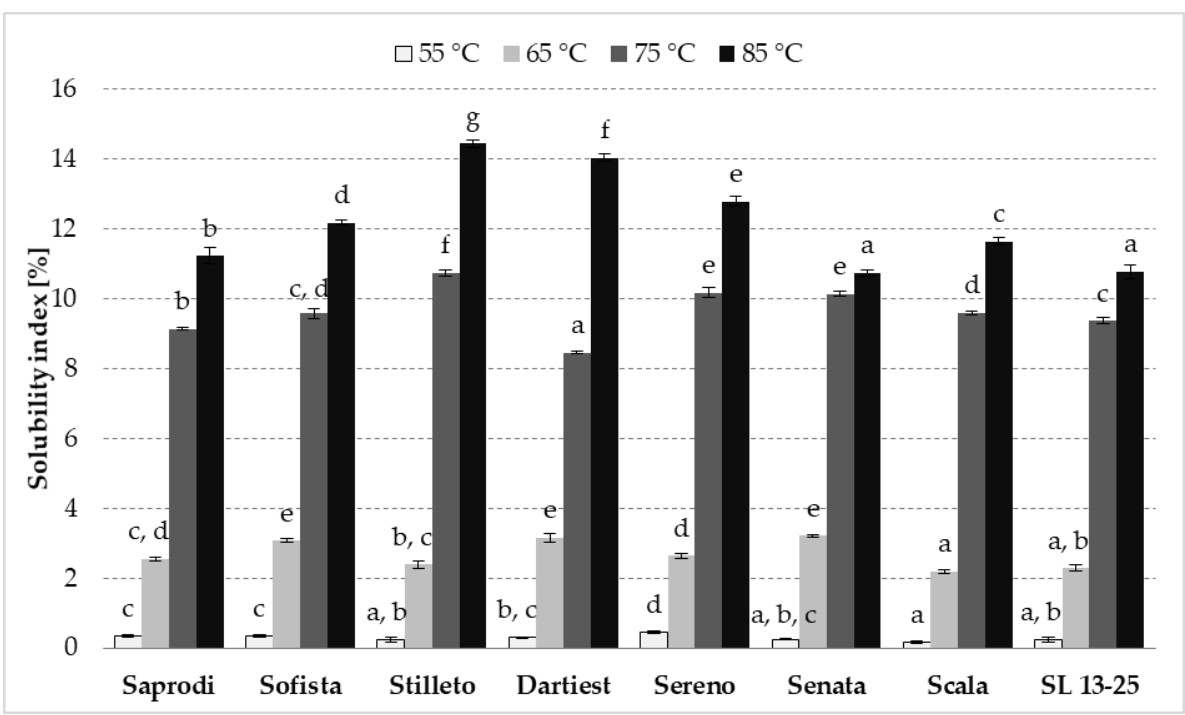

**Figure 2.** Solubility index of the isolated potato starches at different temperatures. Different letters above the bars indicate that the values are significantly different at $p < 0.05$.

### 3.6. Clarity of Starch Paste

Starch paste clarity is affected by the amylose content, molecular weight and starch granule structure, which directly affects starch granule swelling [23]. The transparency data, expressed as % of transmission at 650 nm, are shown in Figure 3. The starch of the Dartiest cultivar had the highest clarity value, followed by the Senata cultivar, while the other cultivars had lower and similar clarity values. The Dartiest cultivar had the lowest proportion of mineral substances and the highest proportion of starch, which is a possible reason for the high clarity.

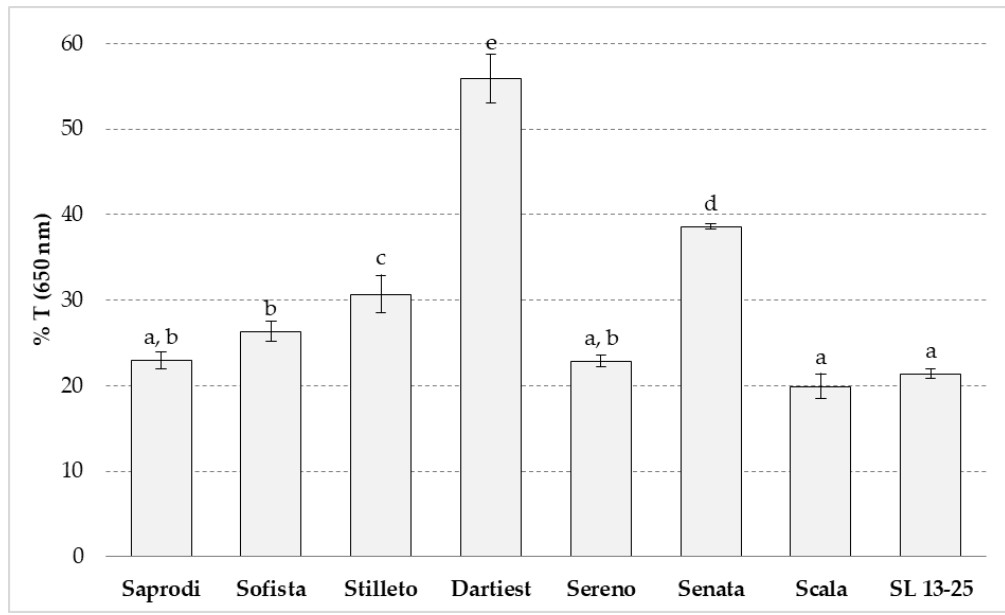

**Figure 3.** Paste clarity (% T) of the isolated potato starches. Different letters above the bars indicate that the values are significantly different at $p < 0.05$.

With gelatinization, starch granules swell and dissociate and let more light through—i.e., their clarity increases [24]. The differences in the starch clarity values may be due to several factors such as the amylose content, lipid and protein content, granule particle size, amylose and amylopectin chain length [22] and phosphate monoester content [25,26].

### 3.7. Texture of Starch Gels

The results of the texture analysis of the starch gels are shown in Table 5. It is known that the strength of the starch gel depends on the size of the "net" formed by the dissolved amylose molecules and on the deformation property of the dissolved starch granules [27]. The Dartiest cultivar had the highest, while the Sereno cultivar had the lowest gel strength. Higher breaking force values were observed for the starch gels of the Sereno and Stilleto cultivars compared to the Dartiest and Senata cultivars. Adhesion refers to the stickiness of the gel with the contact surface. From the results shown in Table 5, it is evident that the Dartiest cultivar had significantly lower adhesion values than the other starch samples. The values for the gel strength are lower than in the study by Sandhu et al. [28], and higher than in the study by Gałkowska et al. [29].

**Table 5.** Texture properties of starch gels of the isolated potato starches.

| Sample | Gel Strength [g] | Rupture Strength [g] | Adhesiveness [g sec] |
|---|---|---|---|
| Saprodi | 3.14 ± 0.08 [b,c] | 928.89 ± 18.84 [d] | −219.78 ± 6.00 [b,c] |
| Sofista | 2.98 ± 0.01 [b] | 840.18 ± 17.90 [b] | −202.52 ± 4.58 [d] |
| Stilleto | 3.25 ± 0.08 [b,c] | 959.33 ± 6.77 [e] | −224.69 ± 4.04 [b] |
| Dartiest | 3.30 ± 0.16 [c] | 776.63 ± 9.16 [a] | −73.02 ± 9.50 [e] |
| Sereno | 2.48 ± 0.23 [a] | 998.99 ± 10.61 [f] | −348.49 ± 2.23 [a] |
| Senata | 3.08 ± 0.00 [b,c] | 758.10 ± 4.46 [a] | −219.27 ± 8.41 [b,c] |
| Scala | 3.03 ± 0.08 [b,c] | 912.11 ± 17.51 [d] | −192.56 ± 5.68 [d] |
| SL 13–25 | 3.08 ± 0.00 [b,c] | 873.06 ± 1.98 [c] | −205.13 ± 8.11 [c,d] |

Values with different letters in the same column are significantly different at $p < 0.05$.

### 3.8. Determination of Thermophysical Properties

3.8.1. Properties of Gelatinization

Gelatinization is an important functional property of potato starch. This process is required to produce the desired functionality, such as the thickening and swelling of potato starch in food products.

Table 6 shows the gelatinization parameters, determined through DSC. The onset temperature of gelatinization ranged from 63.73 °C for the starch obtained from the Dartiest cultivar to 69.18 °C for the starch obtained from the SL 13–25 cultivar. The gelatinization peak temperatures ranged from 66.21 °C for the starch obtained from the Dartiest cultivar to 71.89 °C for the starch obtained from the Senata cultivar, and the final gelatinization temperature ranged from 70.24 °C for the starch obtained from the Dartiest cultivar up to 77.05 °C for the starch obtained from the Scala cultivar. The starch isolated from the Dartiest cultivar had the lowest gelatinization temperature, while the highest gelatinization temperatures were observed in the starches from the SL 13–25, Senata and Scala cultivars. The gelatinization temperatures of the isolated starches are higher than the gelatinization temperatures of the starches obtained from autochthonous potatoes of the Andean region [11], as well as from different potato starches from India [30]. The differences between the transition temperatures of the gelatinization of the starches obtained from different potato cultivars can be attributed to differences in the degree of crystallinity. A high degree of crystallinity enables a stable structure of the granule and makes it more resistant to gelatinization, which is why the gelatinization temperatures are higher [31]. Higher gelatinization temperatures can also be caused by a high double helical arrangement between amylose and amylopectin chains and strong interactions between amylose–amylose chains and amylopectin–amylopectin chains [11]. The gelatinization enthalpies of the potato starches are significantly lower

compared to the enthalpies of the starches obtained from indigenous potato cultivars of the Andean region, and are also different from potato starches from India [11,30].

**Table 6.** Gelatinization parameters of suspensions of the isolated potato starches.

| Sample | $T_o$ (°C) | $T_p$ (°C) | $T_e$ (°C) | $\Delta H$ (J/g) |
|---|---|---|---|---|
| Saprodi | 64.80 ± 0.11 [b] | 70.32 ± 0.36 [d] | 74.24 ± 0.02 [c,d] | 2.16 ± 0.04 [c] |
| Sofista | 63.80 ± 0.18 [a] | 67.56 ± 0.28 [b] | 73.80 ± 0.13 [b] | 1.22 ± 0.06 [a] |
| Stilleto | 64.22 ± 0.42 [a,b] | 68.27 ± 0.33 [c] | 76.73 ± 0.21 [f] | 1.15 ± 0.04 [a] |
| Dartiest | 63.73 ± 0.16 [a] | 66.21 ± 0.13 [a] | 70.24 ± 0.08 [a] | 2.27 ± 0.08 [c,d] |
| Sereno | 67.08 ± 0.06 [c] | 70.11 ± 0.06 [d] | 74.62 ± 0.25 [d] | 2.32 ± 0.05 [d] |
| Senata | 68.45 ± 0.28 [d] | 71.89 ± 0.12 [e] | 75.94 ± 0.08 [e] | 2.01 ± 0.08 [b] |
| Scala | 67.52 ± 0.59 [c] | 71.41 ± 0.06 [e] | 77.05 ± 0.17 [f] | 2.30 ± 0.04 [c,d] |
| SL 13–25 | 69.18 ± 0.15 [e] | 71.84 ± 0.17 [e] | 75.81 ± 0.49 [e] | 2.21 ± 0.02 [c,d] |

Values with different letters in the same column are significantly different at $p < 0.05$. Gelatinization parameters: $T_0$—onset temperature; $T_p$—peak temperature; $T_e$—endset temperature; $\Delta H$—gelatinization enthalpy.

### 3.8.2. Properties of Retrogradation

Retrogradation is a second important functional property of starch, and it is related to the stability of the starch paste during storage. Table 7 shows the values of the initial temperature ($T_0$), peak temperature ($T_p$), end temperature ($T_e$) and retrogradation enthalpy ($\Delta H$) of the isolated potato starch gels after 14 days of storage at 4 °C. The starch gel of the SL 13–25 cultivar had the highest initial temperature, and the Senata cultivar starch gel had the lowest. The starch gel of the Stilleto cultivar had the highest peak temperature, and the starch gel of the Senata cultivar had the lowest. The Dartiest starch gel had the highest final temperature, and the Saprodi starch gel had the lowest. The retrogradation enthalpy values ranged from 0.11 J/g for the Sereno cultivar starch gel to 0.42 J/g for the SL 13–25 cultivar starch gel. The transition temperatures, as well as the enthalpies of retrogradation at the end of the storage period, were significantly reduced compared to the transition temperatures and enthalpies during gelatinization. The results are consistent with the results of Karim et al. [25], where the transition temperatures and retrogradation enthalpies are lower compared to the gelatinization temperatures. After gelatinization, there is a reconnection of starch molecules and their recrystallization, whereby they form less ordered structures, and therefore less energy is needed to melt the restructured crystal in the retrograded starch [25].

**Table 7.** Retrogradation parameters of gels of the isolated potato starches after 14 days of storage at 4 °C.

| Sample | $T_o$ (°C) | $T_p$ (°C) | $T_e$ (°C) | $\Delta H$ (J/g) |
|---|---|---|---|---|
| Saprodi | 44.57 ± 0.43 [b] | 53.11 ± 0.49 [d] | 58.89 ± 0.12 [a] | 0.38 ± 0.00 [c,d] |
| Sofista | 44.80 ± 0.27 [b] | 51.23 ± 0.24 [a,b] | 62.06 ± 0.07 [b,c] | 0.35 ± 0.02 [c] |
| Stilleto | 44.76 ± 0.26 [b] | 55.56 ± 0.59 [e] | 63.19 ± 0.17 [d] | 0.28 ± 0.01 [b] |
| Dartiest | 44.39 ± 0.70 [b] | 55.11 ± 0.04 [e] | 63.27 ± 0.29 [d] | 0.29 ± 0.01 [b] |
| Sereno | 44.70 ± 0.13 [b] | 53.39 ± 0.24 [d] | 62.23 ± 0.28 [c] | 0.11 ± 0.04 [a] |
| Senata | 42.40 ± 0.39 [a] | 50.72 ± 0.35 [a] | 61.53 ± 0.56 [b] | 0.36 ± 0.01 [c] |
| Scala | 45.27 ± 0.42 [b] | 51.99 ± 0.13 [b,c] | 63.09 ± 0.21 [d] | 0.15 ± 0.01 [a] |
| SL 13–25 | 49.19 ± 0.08 [c] | 52.64 ± 0.47 [c,d] | 62.25 ± 0.08 [c] | 0.42 ± 0.03 [d] |

Values with different letters in the same column are significantly different at $p < 0.05$. Retrogradation parameters: $T_0$—onset temperature; $T_p$—peak temperature; $T_e$—endset temperature; $\Delta H$—retrogradation enthalpy.

## 4. Conclusions

The highest dry matter content was found in the Stiletto, while the lowest dry matter content was in the Senata cultivar, and the lowest starch content was found in the SL 13–25 cultivar. The isolated starches from all the cultivars had a high level of purity, which is evident from the analysis of the basic chemical composition of the obtained starches,

where the content of protein, fat, ash and crude fiber is relatively low. The starch of the Saprodi cultivar had the highest and the starch of the Senata cultivar had the lowest value of the L* parameter, although the differences between the values of the L* parameter were very small, so the difference in the color of the starches isolated from different potato cultivars is difficult to see. The starch of the Senata cultivar had the highest color saturation, and the starch of the SL 13–25 cultivar had the lowest. Starch from the Saprodi cultivar had the highest, while starch from the SL 13–25 cultivar had the lowest amylose content. The Dartiest cultivar had the highest, while the Sereno cultivar had the lowest gel strength. The starch gels of the Sereno and Stilleto cultivars had higher breaking force values compared to the Dartiest and Senata cultivars. With an increase in temperature, an increase in the swelling capacity was found in all samples, and a proportional increase in the solubility index with an increase in temperature was also recorded. The starch of the Dartiest cultivar had the highest clarity value, followed by the Senata cultivar, while the other cultivars had lower and similar clarity values. The Dartiest starch had the lowest transitional gelatinization temperatures, and the starches obtained from the SL 13–25, Senata and Scala cultivars showed the highest gelatinization temperatures. The transition temperatures and enthalpies of retrogradation were lower compared to gelatinization. The highest transitional retrogradation temperatures were obtained by the starch gels of the SL 13–25, Stilleto and Dartiest cultivars, while the lowest values were found in the starch gels of the Senata and Saprodi cultivars.

**Author Contributions:** Conceptualization, A.J., D.Š., T.V. and M.K.; methodology, A.J., M.K., T.V. and B.R.; formal analysis, M.K., A.J. and A.L.; investigation, Đ.A., J.B., D.Š. and V.O.B.; writing—original draft preparation, M.K., A.J. and T.V.; writing—review and editing, A.Š., Đ.A., J.B., A.L., B.R. and V.O.B.; supervision, D.Š., T.V. and A.J. All authors have read and agreed to the published version of the manuscript.

**Funding:** This research was funded by Osijek-Baranja County under the project "Isolation and characterization of potato starch produced in the Osijek-Baranja County".

**Institutional Review Board Statement:** Not applicable.

**Informed Consent Statement:** Not applicable.

**Data Availability Statement:** The data presented in this study are available on request from the corresponding author.

**Acknowledgments:** We want to acknowledge to Royal ZAP/Semagri Holland BV for the donation of potato cultivars.

**Conflicts of Interest:** The authors declare no conflicts of interest.

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
