# Peer review of "Isolation and Characterization of Starch from Different Potato Cultivars Grown in Croatia"

_applsci, doi:10.3390/app14020909_

Round 1
Reviewer 1 Report
Comments and Suggestions for Authors
Is the research design appropriate? In my opinion it should be more than one year experiment and simply biochemical assays should be made.
Page 2 row 56 Are you sure the SL 13-25 is cultivar not breeding clone while registration? Regular cultivar will be soon and change name.
Page 2 row 63 What does (32 x 70 cm) is?
Page 2 row 66 How long were potato tubers at 4C degree? "Until laboratory analysis" was one month or more? This was time of tuber maturation while storage. Usually is three months.
Page 2 row 90 What is starch maturation? I can not find any references
Page 2 rows 93-96 Could you define method what you used more than ISO standards? for example: Does the ISO 5983-2 is the Kjeldahl method?
Page 3 row 125 What kind of starch modification were you used?
I suggest reference [10] changing for sth in English
Page 5 row 199 is "table 1" for me should be "Table 1"
name of Table 2 could be Chemical composition of potato starch obtained from the tested probes
This valuable results could be completed to Amylose/Amylopectin ratio. It is Biochemical method from Megazyme.
Comments on the Quality of English LanguageMinor editing of English language required: I found in page 2 row 61 "seed potatoes" for me it should be "potato seeds"
Reviewer 2 Report
Comments and Suggestions for Authors
Overall, the data of this paper is not substantial and innovative enough, has some important defects, and does not meet the journal's desired standard.
The manuscript in its present version is not apposite for publication, and I suggest that this paper be rejected.
Reviewer 3 Report
Comments and Suggestions for Authors
1. In the Abstract, the description of the research background needs to be condensed, and the results and conclusions of the research need to be presented.
2. The article repeatedly mentions the influence of the content of amylose and amylopectin on the properties of starch(Page 6, line 240-242; Page 7, line 264-267; Page 7, line 275-279). What is the reason for not conducting relevant testing?
3. Suggest adding significance analysis to all Figures in this article.
4. Suggest marking a, b, c... in order of average value from high to low.
5. "Properties of gelatinisation" can be changed to subheadings like "3.8.1 Properties of gelatinisation". (Page 10, line 327 )
Round 2
Reviewer 2 Report
Comments and Suggestions for Authors
Overall, the data of this paper is relatively substantial and the analysis is proper. The authors have made sufficient modifications according to the modification comments. The manuscript in its present version is apposite for publication, and I suggest that this paper be accepted without further modification.
Reviewer 3 Report
Comments and Suggestions for Authors
Too many words in the abstract.
Please summarize the research background in 1-2 sentences, then introduce the research methods and content, and finally briefly describe the main conclusions of this article. The abstract section should be kept within 300.
